# Factors Affecting the Use of Pain-Coping Strategies in Individuals with Cerebral Palsy and Individuals with Typical Development

**DOI:** 10.3390/children10010131

**Published:** 2023-01-09

**Authors:** Inmaculada Riquelme, Pedro Montoya

**Affiliations:** 1University Institute of Health Sciences Research (IUNICS-IdISBa), University of the Balearic Islands, 07122 Palma de Mallorca, Spain; 2Department of Nursing and Physiotherapy, University of the Balearic Islands, 07122 Palma de Mallorca, Spain

**Keywords:** pain, pain coping, cerebral palsy, typical development, health, emotion, speech, cognition

## Abstract

Many individuals with cerebral palsy (CP) suffer from pain and must develop pain-coping strategies, although the factors determining them are unknown. This observational study aims at exploring the association between different pain-coping strategies and factors such as age, sex, pain, health status, sleep or motor and cognitive function in individuals with cerebral palsy (CP) and typically developing peers (TD). Main caregivers of 94 individuals with CP (age range = 6–69 years, mean age = 17.78 (10.05)) and the closest relative of 145 individuals with TD (age range = 6–51 years, mean age = 19.13 (12.87)) completed questionnaires on the previous topics (Parent Report of the PEDsQL Pediatric Coping Inventory, the Health Utility Index HUI-3, Epworth Sleepiness Score and the Pittsburgh Sleep Quality Index). Pain presence, duration, intensity, location and ratings of current and worst pain in the last week in an 11-point numerical rating scale were assessed in an interview. Global health was the best predictor the of use of any type of pain-coping strategy, including cognitive self-instruction, problem-solving, distraction, seeking social support and catastrophizing, in both individuals with CP and individuals with TD. However, different health attributes predicted their use in each population. Emotional health was the best predictor in individuals with CP, whereas cognition and pain were the best predictors in individuals with TD. Speech ability was a predictor in both groups. In conclusion, the assessment of health attributes such as emotional health and speech may help design specific interventions for enhancing self-efficacy and adaptive pain coping skills.

## 1. Introduction

More than half of individuals with cerebral palsy (CP) have pain from moderate to severe intensity, on a daily–weekly basis, in multiple body locations [1,2]. Pain can be elicited by comorbidities (muscle spasm, musculoskeletal deformities, gastro-esophageal reflux…) [1] or therapeutic interventions (surgery, poorly fitting equipment, rehabilitation techniques…) [3]. Pain interferes with daily activity [1,4], affecting social roles and reducing participation [1,4] and quality of life in individuals with CP [2].

Adults with CP describe pain as a personal experience rarely discussed with their social or health environment and, consequently, implying little use for analgesia [4]. Furthermore, the effectiveness of pain–relief therapies is controversial [2,3,4], with low levels of satisfaction [5]. In this context, dealing with pain may rely on individual strategies [6]. Adults with CP reported strategies, such as scheduling and pacing daily activities, mind-body disassociation and perseverance, when coping with pain [7], whereas youth with CP described personal factors, such as body image, independence and pain coping, as essential for dealing with their condition [6].

Cognitive–behavioral models postulate that pain coping style plays a crucial role in pain consequences. These models seem equally suitable for individuals with CP, who show consistent links between pain-coping strategies and adjustment to chronic pain [8]. Thus, catastrophizing, seeking solicitous social support or contingent rest are related to maladaptive pain adjustment, higher depression and pain-related disability in individuals with CP, whereas task persistence is associated with less pain interference and better psychological function [4,8,9]. Although these findings have substantiated cognitive–behavioral interventions aimed at enhancing adaptive pain-coping strategies in individuals with CP [8], factors determining the use of one specific strategy are unclear.

Several factors, such as cognitive level, pain intensity, function, subjective health, sleep deprivation and age, are determinants in the use of different pain-coping strategies in populations with chronic pain conditions [10,11,12,13]. Assuming the suitability of the cognitive–behavioral models of pain coping for individuals with CP, we hypothesize that these factors could also affect the use of pain-coping strategies in these individuals. Although recent findings revealed that the use of cognitive strategies in children with CP develops at older ages [9], no study has explored the influence of the previous factors in the use of pain-coping strategies in individuals with CP. This analysis would help detect individuals at risk of using maladaptive strategies and tailor specific interventions. This study aims at exploring the association between different pain-coping strategies and factors, such as age, sex, pain characteristics, global health, sleep disturbance, or motor and cognitive function, in individuals with CP and typically developing (TD) individuals.

## 2. Materials and Methods

### 2.1. Participants

One hundred fifty individuals with CP and 150 TD individuals from Majorca (Spain) were initially contacted through a letter explaining the details of the study. Physiotherapists from specialized centers (schools, secondary schools, occupational centers, day centers and residences) identified participants with CP; simultaneously age-matched TD participants were recruited by asking for volunteers in educational centers: primary schools, secondary schools and universities. Inclusion criteria were the diagnosis of cerebral palsy and ages older than 6 years; the presence of chronic pain was not considered an inclusion criterion. Ninety-four individuals with CP (age range = 6–69 years., mean age = 17.78 (10.05), 34 females) and 145 individuals with typical development (age range = 6–51 years, mean age = 19.13 (12.87), 89 females) agreed to participate in the study. Adults provided written informed consent. For participants under the age of 18, with low cognitive levels or who were legally incapable, parents or legal tutors signed informed consent and participants expressed verbal willingness to participate. This study was approved by the Research Ethics Committee of the University of the Balearic Islands, Spain (ref. 127CER19).

### 2.2. Interview and Questionnaires

Parents or the closest relative in participants with CP, and the closest relative in TD participants, underwent a semi-structured interview consisting of questions on demographic data and pain characteristics. Moreover, they completed several questionnaires on coping strategies, global health and sleep disturbance. To avoid a bias between proxy-reports and self-reports, this procedure was performed for all participants, even if they could self-report. In addition, for participants with CP, the type of cerebral palsy, cognitive level (according to the psychological reports of the center) and level of motor impairment determined by the Gross Motor Function Classification System were obtained from their health history. Table 1 displays the clinical characteristics of participants with CP.

Pain was measured by using the following information from the interview: (1) whether they were experiencing chronic pain or not (pain lasting more than 3 months, yes/no response) and duration of pain; (2) ratings of current and worst pain in the last week by using an 11-point numerical rating scale (0 = no pain, 10 = unbearable pain); and (3) the location of painful body regions by using a drawing of the human figure and pain intensity ratings at each location by using a four-point numerical scale (0 = no pain, 1 = mild, 2 = moderate, 3 = severe) (QL07/00 Pediatric Pain Questionnaire, Spanish version [14]). Pain intensity ratings in all locations were averaged to obtain average pain intensity. Pain scores were set to 0 in all participants with no pain. These interviews, questionnaires and procedures have been previously used in studies of our lab with individuals with CP [2].

Pain-coping strategies were assessed with the Parent Report of the Pediatric Pain Coping Inventory (PPCI, Spanish version) [15] The PPCI consists of 41 items which assess how often different coping strategies have been used on a three-point scale (0 = never, 1 = sometimes, 2 = often). The coping strategies of the PPCI are grouped into the following categories: cognitive self-instruction, problem-solving, distraction, seeking social support and catastrophizing/helplessness. Higher scores indicate more frequent use of a particular coping strategy. This instrument has been previously used in the assessment of pain-coping strategies in individuals with CP [16]. Although this is an instrument designed for children and adolescents, we decided to use it also in the adult population in order to be able to explore potential age-related changes.

Health status was assessed with the Spanish version of the Health Utility Index (HUI-3) [17,18]. The HUI-3 is a widely used classification system for describing the following eight attributes of health: vision, hearing, speech, pain, ambulation, dexterity, emotion and cognition. Higher scores indicate better health in the attribute. Scores for the single attributes and a HUI total score, expressing general health, were calculated. The HUI-3 has been extensively used in the description of health status in individuals with CP [19].

Sleep disturbance was assessed with the Spanish versions of two different questionnaires: the Epworth Sleepiness Scale [20,21] and the Pittsburgh Sleep Quality Index [22,23]. The Epworth Sleepiness Score [20] consists of eight items about the probability of sleeping in daytime situations by using a four-point scale (0 = never, 3 = high chance of dozing). The total score may range from 0 to 24, with scores between 3 and 9 indicating normality, and higher scores indicating mild (10–13), moderate (14–19) and severe (20–24) sleepiness. The Pittsburgh Sleep Quality Index [22] measures sleep quality and disturbances in a 1 month time span by using a four-point scale (0 = never, 3 = three or more times per week). The total score of the scale was computed. Higher scores indicate greater sleep difficulty; scores higher than 5 indicate poor sleep. Both scales have been extensively used in the assessment of sleep disturbances in individuals with CP [24].

### 2.3. Statistical Analyses

Analysis of variance (ANOVA) was performed, including the between-subject factors GROUP (individuals with CP vs. TD individuals) and PAIN (chronic pain vs. non-chronic pain) for the dependent variables pain duration, pain intensity, number of pain locations, coping strategies, health attributes, sleepiness and somnolence. ANOVA results were adjusted by using Bonferroni corrections for post hoc comparisons. Pearson and Spearman correlations, chi-squares and linear standard multiple regression analyses were performed to establish the predictors of coping strategies. Linear multiple regressions were performed separately for individuals with CP and individuals with TD, with the different factors that showed significant correlations with pain coping as predictor variables and the different pain-coping strategies as dependent variables. Durbin–Watson tests were used to check the absence of systematic errors in the multiple regression (all values <2.04 and >1.73). Significant levels were set at *p* < 0.05.

## 3. Results

### 3.1. Differences between Individuals with Cerebral Palsy and Typically Developing Individuals

The presence of chronic pain was higher in individuals with CP than in TD controls (57.4% of individuals with CP, 49.0% of TD individuals, Chi-square = 6.59, *p* = 0.01). The main effect of GROUP (F (1,89) = 4.72, *p* = 0.031) showed longer pain duration in individuals with CP compared with TD controls. The main effects of PAIN were observed in current pain (F (1,209) = 30.75, *p* < 0.001), worst pain this week (F (1,206) = 34.27, *p* < 0.001) and average pain intensity (F (1,219) = 114.43, *p* < 0.001), with higher pain intensity in individuals suffering from chronic pain than in individuals with non-pain. An interaction effect of GROUP x PAIN was found on the number of painful locations (F (1,221) = 7.74, *p* = 0.006), indicating that, in both groups, people with pain had more painful locations than individuals without chronic pain (both *p* < 0.001) and that individuals with CP and chronic pain reported a greater number of painful body locations than TD controls with chronic pain (*p* < 0.001), whereas no differences were revealed in individuals without pain (*p* = 0.382).

Table 2 displays the descriptive statistics of the frequency of use of coping strategies for each group. A main effect of GROUP was found in all coping strategies (all F > 14.08, all *p* < 0.001), revealing that individuals with CP used them more frequently compared to TD controls. No significant effects due to PAIN or to the interaction of GROUP x PAIN were found.

An interaction effect was observed on the HUI-3 score of the pain attribute (F (1,165) = 3.93, *p* = 0.049), indicating that there were no significant differences between individuals with CP or TD if they had pain (*p* = 0.734), but in the absence of chronic pain, individuals with CP had lower scores than in TD controls without chronic pain (*p* = 0.025); moreover, no significant differences in the pain attribute between individuals with or without chronic pain were found for any of the groups (both *p* > 0.072). The rest of the HUI attributes and the HUI total score revealed the main effects of GROUP (all F > 5.35, all *p* < 0.021), with lower scores (poorer health) in individuals with CP than in TD controls. No significant effects were found on the HUI emotion attribute.

There were no sleep disturbances in any of the groups or conditions. A main effect of GROUP was found on sleepiness (F (1,159) = 5.6, *p* = 0.019) with lower sleepiness in individuals with CP than in TD controls.

### 3.2. Predictors of Pain-Coping Strategies in Individuals with Cerebral Palsy and Controls with Typical Development

Multiple regression analyses were performed separately for each group of participants using each of the coping strategies as the dependent variables. Predictor variables were selected from the variables of pain, age and sleep disturbance, and the health status whose correlations with coping strategies were statistically significant (Table 3). The same criteria were followed for gross motor function level, cognitive level and the type of CP in individuals with CP. All pain-coping strategies showed significant correlations with the HUI total score (r range = −0.37/−0.43, all *p* < 0.01). Distraction was correlated with somnolence (*p* < 0.05). Distraction and seeking social support were also significantly correlated with age (r range = −0.14/−0.22, all *p* < 0.05). No significant correlations were found among coping strategies and any of the pain variables, type of cerebral palsy, gross motor function or cognitive level. Chi-square comparisons did not show significant associations among coping variables and sex (all *p* < 0.08). Those variables involved in significant correlations were used as predictors. In addition, the presence of chronic pain was included in the regression analyses to evaluate the effects of chronic pain on pain-coping strategies.

Table 4 displays the statistics of the regression analyses for each group and for the different pain-coping strategies. In TD controls, all pain-coping strategies (except distraction) were significantly predicted by the HUI total score (all R < 0.43, all F > 0.82, all *p* < 0.516). Additionally, seeking social support was predicted by age. In individuals with CP, all pain-coping strategies were predicted by the HUI total score, and catastrophizing/helplessness was also predicted by age (all R < 0.54, all F > 2.56, all *p* < 0.05).

To further examine the role of the HUI attributes for each of the coping strategies in both groups, further regression analyses were conducted using the HUI attributes as predictors (Table 5). The regression analyses showed that different predictors accounted for a significant proportion of the variance in TD controls (all R < 0.50, all F > 0.71, all *p* < 0.686) and individuals with CP (all R < 0.60, all F > 2.61, all *p* < 0.018). In TD controls, cognitive self-instruction was predicted by speech; problem-solving was predicted by pain and cognition; seeking social support was predicted by cognition; catastrophizing/helplessness was predicted by pain and speech. In individuals with CP, all coping strategies were predicted by speech and emotion; problem-solving was also predicted by vision.

## 4. Discussion

The aim of the present study was to explore the association between different pain-coping strategies and several pain-related factors, such as age, sex, pain characteristics, health status, sleep disturbance or motor or cognitive function, in individuals with CP and TD controls. Our results showed that health status was the best predictor for the use of pain-coping strategies in both groups, although different health factors were relevant in each group of individuals. Thus, the use of coping strategies was associated with emotional health in participants with CP; whereas coping was more related to cognitive factors in individuals with TD. Interestingly, the speech ability was relevant for the use of coping strategies in both groups.

Individuals with CP reported a greater presence of chronic pain at more different body locations and with longer duration than controls with TD. Accordingly, individuals with CP also reported more frequent use of all types of pain-coping strategies than participants with typical development. Moreover, the factors affecting pain coping equally modulated the use of active and passive strategies both in individuals with CP and with TD. Thus, our findings seem to better characterize the amount of the use of pain-coping strategies than the quality of them. This greater use of all types of pain-coping strategies has also been reported in other pathologies suffering from chronic pain [25]. However, other studies have associated factors such as poor subjective health or poor mental health with the use of passive pain-coping strategies in individuals with chronic pain conditions [11,26]. Likewise, high pain intensity has been related to higher use of catastrophizing in individuals with CP [27]. The use of some coping strategies, such as catastrophizing or seeking support, has been associated with higher pain-related disability and depression, whereas the use of other strategies, such as task persistence, has resulted in less pain interference and better psychological health in individuals with CP [4,8,9,28]. Further research is warranted for reinforcing the evidence for the use of interventions promoting the learning of appropriate pain-coping strategies in this population [8].

Health status was an important predictor for the use of pain-coping strategies in both populations, indicating that worse health was associated with more frequent use of coping strategies. Illness symptoms have consistently correlated with coping strategies and poor subjective health or unexplained symptoms have been predictors of maladaptive coping strategies in individuals with low back pain [11]. It has been suggested that stress produced by pain can modify the hypothalamic–pituitary–adrenal axis neuroendocrine activity and that sustained cognitive activation can be mediated by coping and health complaints [27]. Accordingly, it appears that patients with chronic pain have more health complaints and negative expectancies and show reduced cortisol reactivity [27]. In agreement with this research, we found that emotional responses to the health condition rather than pain intensity parameters were better predictors of coping strategies and mediated the relationship between coping strategies and physical function in CP. In TD participants, however, a pain score mixing intensity and interference (HUI pain) was the best predictor of several coping strategies, but not in individuals with CP. Further research is warranted to elucidate the best parameters linking pain to pain-coping strategies and to explore the apparent dissociation between pain and pain-coping strategies in individuals with CP.

Poor speech predicted higher use of pain-coping strategies both in individuals with CP and TD controls. Poor verbal ability has been previously linked to an increment of behavioral coping strategies [29]. Self-efficacy for pain communication has been related to lower levels of pain [30] and vocalizing in pain situations increased pain tolerance [31]. Therefore, it seems logical that the inability to use verbal pain-coping strategies, such as emotional verbalization or positive self-verbalization, could lead to the substitution for other strategies to conceptualize pain (e.g., cognitive self-instruction) or develop pain control techniques (e.g., distraction or problem-solving) in individuals with CP. Furthermore, the use of self-talk coping statements and pain discussion has been negatively related to anxiety, depression and catastrophizing in patients with chronic pain [30]. Similarly, in our individuals with typical development, the ability to be understood by strangers or by people who know them well seems to affect pain-coping strategies such as cognitive self-instruction or catastrophizing. Thus, poor speech may increase both adaptive and maladaptive pain-coping strategies and seem to affect equally individuals with CP and TD individuals.

Although cognitive factors and pain intensity may play a relevant role in predicting the use of adaptive and maladaptive coping strategies in TD controls, they are not good predictors for the use of coping skills in CP, as it happens in patients with chronic pain [11]. Furthermore, it seems that emotional factors may play a key role in individuals with CP. Indeed, previous studies have reported that individuals with CP have deficits in emotion regulation, affective expression and externalizing behaviors compared to TD individuals [32]. The ability to self-regulate emotions has been related to emotional impact, adjustment and coping strategies for pain [33]. On the other hand, strategies to self-regulate emotions, such as the capacity of attenuating negative emotions, have been predictors of pain and health outcomes [34]. Therefore, it could be that deficits in emotion regulation in individuals with CP would result in increased use of coping strategies. In addition, the verbal inability to express emotions related to pain could divert the emotional approach to other coping strategies. In this sense, poor mental health has been associated with emotional and avoidance coping strategies [26], whereas good self-efficacy for pain communication has been found essential to self-regulate emotional expression [35]. The combination of speech and emotional deficits may be relevant in the election of pain-coping strategies in individuals with CP.

Age showed significant correlations with several pain-coping strategies, such as distraction, social support and catastrophizing, revealing higher use of pain-coping strategies in lower ages. In addition, age was a predictor of seeking social support in individuals with TD, indicating higher use of this strategy in younger individuals. In individuals with CP, age was a predictor of catastrophizing, with higher use of this strategy at lower ages. This is in accordance with other studies pointing to the use of different coping strategies at different ages in TD individuals with chronic pain [36,37] and showing an age-related development of active strategies at older ages, also in individuals with CP [9]. Although health factors might be more determinant than age as predictors of pain coping, the intervention on coping strategies should incorporate a lifespan–developmental perspective.

Study limitations: Proxy-reports, although allowing the inclusion in the study of individuals with low cognitive levels, might be different from self-reports, especially when caregivers report in participants with severe cognitive and verbal impairments, where proxy reports of pain could be inaccurate; moreover, self-reports would have given a more accurate portrayal of the variables. Questionnaires were not intended or validated across all the age ranges. The cognitive level of TD individuals was not assessed, although the data collection procedure was identical to that used in individuals with CP. Gender differences between the groups, as well as the wide age range or the diverse cognitive level, could have biased the results, although these factors did not predict coping strategies in any of the groups. Verbal abilities or the use of augmentative or alternative communicative devices were not assessed, preventing a deeper explanation of the influence of speech on coping strategies. The low number of participants in each gross motor function level did not allow for analyzing potential differences in coping strategies in mild and severe disabilities; moreover, the high percentage of individuals with CP with severe impairment makes the generalization of the findings difficult. The percentage of chronic pain in the control group was rather high; it is likely that individuals with chronic pain were more motivated to participate as volunteers for a control group in a research studying pain.

In conclusion, the present study seems to point to emotional health status as a relevant factor in predicting the use of coping strategies in individuals with CP, whereas cognitive factors influencing health appear to be more predictive of coping strategies in TD controls. These findings suggest that deficiencies in emotion regulation, together with difficulties in the verbal expression of pain, could affect the use of pain-coping strategies in individuals with CP. Although the design of the study aimed to include individuals with severe impairment, scantily represented in pain studies, this approach had a risk of bias that makes implications unclear and does not allow generalizing the results. However, our findings open a new path to be explored with future research integrating emotion and pain processes in individuals with CP.

## Figures and Tables

**Table 1 children-10-00131-t001:** Clinical characteristics of individuals with cerebral palsy. GMFCS = Gross motor function classification system; describes the gross motor function according to 5 levels: 1 = walks without limitations, 5 = transported in a manual wheelchair.

Participants with Cerebral Palsy	Clinical Variable
Type of cerebral palsy	
Bilateral spastic	61
Unilateral spastic	4
Dyskinetic	19
Ataxic	10
Motor impairment (GMFCS)	
Level I	14
Level II	11
Level III	15
Level IV	13
Level V	41
Cognitive impairment	
None	28
Mild	11
Moderate	13
Severe	42

**Table 2 children-10-00131-t002:** Means and standard deviation of the frequency of pain-coping strategies in individuals with cerebral palsy and controls with typical development.

	Cognitive Self-Instruction	Problem-Solving	Distraction	Seeking Social Support	Catastrophizing
Typically developing controls	0.84 (0.44)	0.91 (0.35)	0.74 (0.40)	0.71 (0.34)	0.72 (0.30)
Individuals with cerebral palsy	1.48 (1.16)	1.27 (0.88)	1.43 (1.17)	1.38 (0.98)	1.23 (0.96)

**Table 3 children-10-00131-t003:** Correlations among pain-coping strategies and potential affecting factors in individuals with cerebral palsy and controls with typical development.

	Cognitive Self-Instruction	Problem-Solving	Distraction	Seeking Social Support	Catastrophizing
Age	0.019	0.024	−0.155 *	−0.227 **	−0.035
Pain duration	−0.027	0.088	−0.024	−0.075	0.085
Current pain intensity	−0.032	0.021	−0.093	−0.110	0.008
Worst pain last week	−0.040	0.126	−0.147	−0.019	−0.018
Average pain intensity	0.022	0.070	0.043	0.023	0.092
Number of painful locations	−0.030	0.067	−0.078	−0.039	0.058
HUI total score	−0.341 **	−0.310 **	−0.371 **	−0.430 **	−0.302 **
Somnolence	−0.047	0.049	−0.125 *	−0.065	−0.017
Sleepiness	−0.026	0.097	−0.072	0.018	0.146
Gross motor function (in CP)	0.043	0.096	0.127	0.052	0.070
Cognitive level (in CP)	0.151	0.183	0.153	0.100	0.131
Type of cerebral palsy (in CP)	0.086	0.104	0.078	0.056	0.083

* *p* < 0.05, ** *p* < 0.01.

**Table 4 children-10-00131-t004:** Multiple regression predictors of pain-coping strategies in individuals with cerebral palsy and controls with typical development.

	Cognitive Self-Instruction			Problem-Solving			Distraction			Seeking Social Support			Catastrophizing		
Predictors	Beta	Adj R2	R2	Beta	Adj R2	R2	Beta	Adj R2	R2	Beta	AdjR2	R2	Beta	Adj R2	R2
Typically developing controls															
Presence of chronic pain	0.11			−0.09			0.09			0.13			−0.09		
Age	0.16			−0.19			−0.10			−0.33 **			−0.13		
Somnolence	−0.02			0.15			0.04			0.10			0.19		
Global health total score	−0.23 *			−0.27 **			−0.13			−0.26 **			−0.26 *		
		0.047	0.086		0.098	0.135		−0.007	0.034		0.153	0.188		0.104	0.140
Individuals with cerebral palsy															
Presence of chronic pain	−0.01			−0.02			−0.02			0.02			−0.09		
Age	−0.21			−0.23			−0.21			−0.18			−0.28 *		
Somnolence	−0.10			−0.15			−0.09			−0.04			−0.13		
Global health total score	−0.36 **			−0.30 *			−0.48 ***			−0.44 **			−0.35 **		
		0.123	0.188		0.103	0.170		0.233	0.290		0.161	0.223		0.161	0.223

* *p* < 0.05, ** *p* < 0.01, *** *p* < 0.001.

**Table 5 children-10-00131-t005:** Multiple regression predictors of pain-coping strategies related to health domains in individuals with cerebral palsy and controls with typical development.

	Cognitive Self-Instruction			Problem-Solving			Distraction			Seeking Social Support			Catastrophizing		
Predictors	Beta	Adj R2	R2	Beta	Adj R2	R2	Beta	Adj R2	R2	Beta	AdjR2	R2	Beta	Adj R2	R2
Typically developing controls															
Vision	0.004			−0.13			−0.01			0.03			0.05		
Hearing	−0.12			0.01			−0.04			−0.06			−0.09		
Speech	−0.31 *			−0.21			−0.29			−0.26			−0.39 **		
Pain	−0.13			−0.23 *			−0.02			−0.05			−0.28 **		
Ambulation	0.14			0.27			−0.04			0.29			0.57		
Dexterity	−0.07			−0.08			0.05			−0.25			−0.37		
Emotion	0.16			−0.01			0.17			0.08			−0.02		
Cognition	−0.20			−0.20 *			−0.07			−0.24 *			0.01		
		0.029	0.108		0.096	0.169		−0.024	0.058		0.026	0.105		0.187	0.252
Individuals with cerebral palsy															
Vision	−0.21			−0.28 *			−0.14			−0.17			−0.16		
Hearing	0.18			0.23			0.12			0.11			0.13		
Speech	−0.67 **			−0.56 *			−0.57 *			−0.66 **			−0.52 *		
Pain	−0.03			0.03			−0.03			−0.01			0.06		
Ambulation	−0.16			−0.18			−0.17			−0.17			−0.08		
Dexterity	0.07			0.14			−0.03			0.07			0.03		
Emotion	−0.34 *			−0.35 *			−0.30 *			−0.36 **			−0.33 *		
Cognition	0.29			0.24			0.18			0.26			0.13		
		0.334	0.225		0.184	0.299		0.250	0.355		0.259	0.263		0.192	0.305

* *p* < 0.05, ** *p* < 0.01.

## Data Availability

The data presented in this study are available on request from the corresponding author.

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
