# Peer review of "Factors Affecting the Use of Pain-Coping Strategies in Individuals with Cerebral Palsy and Individuals with Typical Development"

_children, 2023, doi:10.3390/children10010131_

Round 1
Reviewer 1 Report
Dear authors,
Thank you for your important work! This is a relevant and interesting topic. I do, however, have some major concerns with the article, some of which are highlighted below.
· Title - Use of word “healthy individuals”. Although it is understood that this control group did not have cerebral palsy it does not seem as if any other exclusion criteria were used. How do you know they were “healthy”?
· Abstract – It would be helpful to see the age span of the participants in the abstracts.
· Abstract – Unclear what is meant by the word “disability” in this context.
· Abstract – phrasing line 14 “Main caregivers of 94 CP and the closest relatives of 145 TD”? Also, please define CP the first time it is used. Please use person-first language throughout.
· Abstract – unclear why speech ability would be measured in 145 apparently “healthy” individuals and why/how this would be a parameter that would have any/much variability.
· Introduction – line 38, “spontaneous use of scheduling”. Unclear what is meant here.
· Introduction – line 44 – unclear
· Section on what factors are associated with different pain coping strategies. This section is somewhat confusing as it mixes literature on preterm children, patients with low back pain, patients with fibromyalgia.
· Materials and Methods. It seems as if all ages are included. Coping strategies are clearly different depending on the age of the person. This does not seem to be really reflected in this study. It might have been more informative to focus on a more defined, narrow, age range.
· Materials and Methods. Why was chronic pain defined as pain lasting longer than 6 months? Normally it is defined using a cut point of 3 months. Any particular reason?
· Materials and Methods. Duration of pain is measured in years. This is somewhat crude when anyone over 6 years can be included. Clearly, a 6-year-old will not have had the opportunity to be in pain for as many years as an adult.
· Materials and Methods. Lines 110-113 – Why is this relevant given that self-report was not used?
· Results – Interesting with such a high level of pain in the control group. ‘
· Discussion – given the design of the study, with such a broad age range, a relatively small number of individuals per GMFCS level for instance, the implications of the study are unclear.
Reviewer 2 Report
This is a cross-sectional descriptive study in which information investigating predictors of the coping strategies used to deal with pain by children and adults with cerebral palsy (CP) as well as their age-matched peers without CP.
The topic is an important one, as pain is under-researched and under-treated in CP. The sample size is good, and the researchers have collected a considerable amount of data by interview from participants.
However, there are two significant problems with the paper.
1. The analyses are not hypothesis-driven. A great many potential predictor variables are included in the correlations and multiple regression to choose which ones predict coping strategies. Decisions are made on statistical grounds not theoretical grounds: e.g., “Predictor variables were selected from the variables of pain, age, and sleep disturbance and HUI whose correlations with coping strategies were statistically significant.”
2. The discussion and conclusions focus on predictors of how much coping strategies are reported as being used, without distinguishing between the different types of coping strategies. The authors recognize that some coping strategies are more adaptive than others, and cite literature to show that other populations tend to use adaptive or maladaptive strategies more under certain conditions. But in presenting their own data, they focus too little on the types of coping strategies used.
Other issues:
1. There were only 2 inclusion criteria: CP and age > 6. Many of the individuals with CP were adults. However, all participants in the study had “parents or the relatives providing the major care”. Many adults with CP in the community do not live with their parents or relatives providing major care. So were there stricter criteria for the adults? It is notable that nearly half of the people with CP were classified as GMFCS V, which is much higher than in the CP population in most countries (Reid et al 2011 Dev Med Child Neurol. Nov;53(11):1007-12).
2. How valid is the PPCI for this population? It was designed for 5- to 16-year-old children. In this study, it is applied to adults. Moreover, nearly half of participants (42/94) have a “severe cognitive impairment” and presumably a large proportion were non-verbal (as 41/94 were GMFCS V); and so is it possible that all caregivers could accurately know what coping strategies are being used (particularly the more cognitive ones, such as self-instruction, problem solving, and catastrophizing)?
3. Were Spanish versions of the tools used?
4. Please include details of the Ethics approval obtained for this research.
5. Under statistical analysis, please mention the dependent variables for the ANOVAs and the type of multiple regression used.
6. In the Results, where interaction effects are significant, please interpret the interactions. (At present, they are interpreted as if they were main effects.)
7. If you decide to include correlations, please provide a table of these.
8. Tables 3 and 4 are landscaped tables printed on portrait page orientation, so the right-most columns are cut off.
9. In this study, greater impairment of verbal skills is associated with greater use of pain coping strategies in this study. Did these participants have more severe impairments overall? Did they have more pain? Might one explanation of this and other findings be that individuals with more pain use more of coping strategies, while those with little or no pain don’t need any coping strategies, so their PPCI scores are naturally lower?
10. The most recent reference in the reference list is 2019. There have been more recent papers published since then on coping with pain in CP. Please consider whether you want to include any of these.
There are some minor issues:
1. In the abstract, “CP” à “cerebral palsy (CP)”
2. “more of a half” à “more than a half”
3. “social roles1” à “social roles”
4. “pain evolution” Please clarify what you mean here.
5. “sleep deprivation merged catastrophizing in patients with fibromyalgia” Please express more clearly. I think “merged” is not the word you want.
6. “no study have explored” à “no study has explored”
7. “individuals in risk” à “individuals at risk”
8. “Adults with preserved cognition” Please rephrase
9. “juridical tutored” Please clarify.
10. “semi-structure interview” à “semi-structured interview”
11. “were average” à “were averaged”
12. “These interview” à “these interviews”
13. PEDsQL Pediatric Coping Inventory à Pediatric Pain Coping Inventory
14. Epworth Sleepiness Score à Epworth Sleepiness Scale
15. “no sleep disturbances in none of the groups or conditions” à “no sleep disturbances in any of the groups or conditions”
16. “solicitous” Is this really the word you mean?
17. “strategies9” à “strategies”
18. “show to predict” à “predict”
Round 2
Reviewer 1 Report
I do think it is improved. Somebody needs to check language/grammars however.
Author Response
Thanks for your suggestion. We have checked the language.
Reviewer 2 Report
This is the second round of review for this paper. The authors have carefully addressed all of the concerns that I raised in my previous review.
I have a very few suggestions this time:
1. I think the authors should acknowledge that the PPCI was designed for children and adolescents, not adults.
2. In my previous review I asked for the type of multiple regression. I should have stated more clearly what I meant by that question. I meant could you please indicate whether this is standard multiple regression (where the variance reported is unique to each IV) or stepwise (where IVs are entered in order of their contribution to the model) or hierarchical (where you pre-determine the order of entry based on theory) or what. I suspected in my first reading that you had used stepwise, but now I think it’s probably standard multiple regression, and that the variance reported is unique variance due to each IV (which is better than using stepwise). Could you please specify this in the data analysis section?
3. In my previous review, I questioned whether all caregivers could accurately know what coping strategies are being used, especially by the nonverbal participants. The authors have responded by adding in a sentence to the Discussion, acknowledging the possibility that proxy reports may be different from self-reports, especially for those with severe cognitive and verbal impairment. Since my previous review, I have been interviewing parents of children with intellectual disabilities (including but not restricted to CP), and the majority say that they wouldn’t know how much pain their child was experiencing. Many of these children have a high pain threshold, where the child doesn’t react to an injury nearly as much as you would expect, so the parent may not realize that anything is wrong. So I think proxy reports of pain (not only coping strategies) are likely to be inaccurate in this population, under-estimating the amount of pain. The authors may consider adding this to their discussion.
And just a few typos:
Change “adjusment” to “adjustment”
Change “waranted” to “warranted”
Change “significat” to “significant”
Change “specially” to “especially”
Remove “shoul”
Change “scantly” to “scantily”
Author Response
This is the second round of review for this paper. The authors have carefully addressed all of the concerns that I raised in my previous review.
I have a very few suggestions this time:
- I think the authors should acknowledge that the PPCI was designed for children and adolescents, not adults.
We have added a sentence in page 4 (Materials and Methods, 2.2 Interview and questionnaires) acknowledging this fact and explaining that we decided to use it also in the adult population in order to explore potential age-related changes.
- In my previous review I asked for the type of multiple regression. I should have stated more clearly what I meant by that question. I meant could you please indicate whether this is standard multiple regression (where the variance reported is unique to each IV) or stepwise (where IVs are entered in order of their contribution to the model) or hierarchical (where you pre-determine the order of entry based on theory) or what. I suspected in my first reading that you had used stepwise, but now I think it’s probably standard multiple regression, and that the variance reported is unique variance due to each IV (which is better than using stepwise). Could you please specify this in the data analysis section?
We have clarified the type of multiple regression in the section 2.3 Statistical analyses.
- In my previous review, I questioned whether all caregivers could accurately know what coping strategies are being used, especially by the nonverbal participants. The authors have responded by adding in a sentence to the Discussion, acknowledging the possibility that proxy reports may be different from self-reports, especially for those with severe cognitive and verbal impairment. Since my previous review, I have been interviewing parents of children with intellectual disabilities (including but not restricted to CP), and the majority say that they wouldn’t know how much pain their child was experiencing. Many of these children have a high pain threshold, where the child doesn’t react to an injury nearly as much as you would expect, so the parent may not realize that anything is wrong. So I think proxy reports of pain (not only coping strategies) are likely to be inaccurate in this population, under-estimating the amount of pain. The authors may consider adding this to their discussion.
Pain detection in individuals with cerebral palsy is one of our research lines. Our lab has conducted studies comparing pain intensities reported by individuals with CP and proxies’ (their physiotherapists) in different pain situations, or comparing physiotherapists and parents reports. Althought there are situations, mostly related to the general description of presence of pain, pain intensity and interference, where individuals with CP and their proxies had a high agreement, it is true that this agreement was more variable when specifically asking for different health procedures and when it was referred to non-verbal individuals. Thus, although proxies could be able to detect al least the presence and intensity of pain, we agree with the reviewer that this could be more difficult in individuals with more severe cognitive impairment. We have modified the previous sentence in the discussion to specifically cover inaccuracy in pain reports.
Riquelme, I.; Cifre, I.; Montoya, P. Are physiotherapists reliable proxies for the recognition of pain in individuals with cerebral palsy? A cross sectional study. Disabil Health J 2015, 8, 264-70. doi: 10.1016/j.dhjo.2014.08.009.
Riquelme, I.; Pades-Jiménez, A.; Montoya, P. Parents and physiotherapists recognition of non-verbal communication of pain in individuas with cerebral palsy. Health Comm. 2017, 33, 1448-53. https://doi.org/10.1080/10410236.2017.1358243
And just a few typos:
Change “adjusment” to “adjustment”
Change “waranted” to “warranted”
Change “significat” to “significant”
Change “specially” to “especially”
Remove “shoul”
Change “scantly” to “scantily”
Typos have been corrected.
